# Antioxidant Potential and Enhancement of Bioactive Metabolite Production in In Vitro Cultures of *Scutellaria lateriflora* L. by Biotechnological Methods

**DOI:** 10.3390/molecules27031140

**Published:** 2022-02-08

**Authors:** Inga Kwiecień, Natalizia Miceli, Manuela D’Arrigo, Andreana Marino, Halina Ekiert

**Affiliations:** 1Department of Pharmaceutical Botany, Faculty of Pharmacy, Jagiellonian University Medical College, Medyczna 9 Str., 30-688 Krakow, Poland; halina.ekiert@uj.edu.pl; 2Department of Chemical, Biological, Pharmaceutical and Environmental Sciences, University of Messina, Viale F. Stagno d’Alcontres, 31, 98166 Messina, Italy; nmiceli@unime.it (N.M.); manuela.darrigo@unime.it (M.D.); andreana.marino@unime.it (A.M.)

**Keywords:** *Scutellaria lateriflora*, in vitro cultures, *Scutellaria*-specific flavonoids, verbascoside, antioxidant activity, phenylalanine, tyrosine, methyl jasmonate

## Abstract

Studies carried out using three different in vitro assays and a biological setting (*Escherichia coil*) demonstrated the antioxidant activity of *Scutellaria lateriflora* microshoot extract. Moreover, the extract exhibited no toxicity in a brine shrimp lethality bioassay. These results indicated that microshoots are a rich, safe source of antioxidants, which encouraged us to enhance their production in vitro. In agar and agitated cultures, two biotechnological strategies were applied: feeding the cultures with the biogenetic precursors of the phenolics—phenylalanine and tyrosine, and eliciting them with methyl jasmonate. Specific *Scutellaria* flavonoids and verbascoside were analysed by HPLC. Feeding with precursors (1 g/L) in agar cultures decreased the production of the metabolites. In agitated cultures, different concentrations of precursors (1.0–2.5 g/L) and the elicitor (10; 50; 100 µM) were tested. Additionally, parallel feeding with the precursor and elicitor in a concentration of 50 µM were applied. The best strategy for total flavonoid and verbascoside production was phenylalanine feeding (1.5 g/L), max. 3765 and 475 mg/100 g DW, respectively, after 7 days. This is the first report documenting the high antioxidant production in *S. lateriflora* microshoots after feeding with phenylalanine. Moreover, for the first time, bioreactor cultures were successfully maintained, obtaining attractive results (max. total flavonoid content 2348 and verbascoside 485 mg/100 g DW).

## 1. Introduction

Apart from *Scutellaria baicalensis* (Baikal skullcap), the most valuable species of the *Scutellaria* genus is *Scutellaria lateriflora* (American skullcap, blue skullcap) [1]. This species has long been used in traditional medicine by the indigenous people of North America [2]. It still holds an important position in the USA where it is used as a sedative and an anticonvulsant for the treatment of epilepsy, hysteria, and various neurotic ailments [3,4,5,6]. The raw material utilised from this species is the herb [7]. The main components of the *S. lateriflora* herb are flavonoids such as baicalein, baicalin, wogonin, wogonoside, scutellarein, scutellarin, and oroxylin A, which are specific for the genus *Scutellaria*. In addition, the raw material contains also other flavonoid compounds that are commonly found in the plant kingdom, including luteolin or apigenin, simple phenolic acids (caffeic, p-coumaric, ferulic, and protocatechuic acids), depsides (e.g. chlorogenic acid), iridoid compounds (e.g., catalpol), phenylpropanoid glycosides (e.g., verbascoside), and diterpenoid-neoclerodane derivatives [1,6,8,9,10].

The plant raw materials used in the herbal industry are mostly obtained from wild plants harvested from natural locations and from cultivated plants. The quality of such raw materials varies to a great extent, and hence, possess a significant challenge to obtaining high-quality standardised products. However, this can be overcome by in vitro cultures and plant biotechnology techniques. In vitro cultures allow the rapid cultivation of a large amount of homogeneous, highly productive biomass. The type of culture and chemical composition of the medium are the main factors influencing the production of secondary metabolites in biomass cultured in vitro [11]. Advanced strategies such as precursor feeding, elicitation by biotic and abiotic elicitors, selection of high-productive cell lines, and genetic transformation have been applied for stimulating the in vitro production of metabolites [12,13,14].

Biotechnological studies conducted on *S. lateriflora* have mainly focused on analyzing the usefulness of “hairy root” cultures, resulting from the genetic transformation of this species with *Rhizobium rhizogenes* (*Agrobacterium rhizogenes*) as a source of secondary metabolites [15,16,17], and on the micropropagation protocols [18,19]. The results of our previous study indicated that *S. lateriflora* microshoot culture is a rich source of bioactive metabolites and can serve as a valuable model for biotechnological experiments. We have already assessed the influence of the composition of basal media (Murashige and Skoog (MS), and Linsmaier and Skoog (LS)) and the concentration of selected plant growth regulators (PGRs)-6-benzyladenine (BA) and 1-naphthaleneacetic acid (NAA) on the accumulation of flavonoids, phenylpropanoid glycosides, and phenolic acids in two types of cultures (agar and agitated). We documented that the concentration of PGRs in the MS and LS media had an impact on the growth and biosynthetic potential of *S. lateriflora* cells cultured in vitro. The MS media favoured biomass growth, while the LS media favoured metabolite production. Compared to agitated cultures, the total content of *Scutellaria*-specific flavonoids and verbascoside was higher in agar cultures (max. 2986 and 543 mg/100 g dry weight (DW), respectively) [20]. We also investigated the effect of light conditions (monochromatic red and blue light, white light, UV-irradiation, and darkness) on the accumulation of secondary metabolites in agar cultures, and found that blue light had the most beneficial effect on metabolite accumulation [21]. Our previous results from the analysis of the biotransformation process also demonstrated that agitated microshoot cultures had high biosynthetic potential. Using an exogenous substrate (hydroquinone) at a concentration of 288 mg/L, we obtained the maximum content of 5.63 g/100 g DW of arbutin (hydroquinone glucoside) as the result of the β-D-glucosilation process [20].

One of the goals of the presented research was to evaluate the antioxidant potential of extract obtained from *Scutellaria lateriflora* microshoots cultured in vitro using four different methods, and its potential safety using the brine shrimp lethality bioassay. Moreover, in this study, we analysed whether the in vitro cultures of *S. lateriflora* can serve as an alternative source of biologically active secondary metabolites such as flavonoids and verbascoside (phenylpropanoid glycoside, disaccharide caffeoyl ester)—compounds with high antioxidant activity. Based on our previous results on *S. baicalensis* [22], we fed the culture medium with biosynthetic precursors and stimulated it by elicitation. The medium was supplemented alone and parallel with precursors and the elicitor. The experiment was carried out in agar and agitated cultures. Aromatic amino acids, namely phenylalanine and tyrosine, were used as precursors of phenolic compounds. Metabolites with a phenolic structure are biogenetically related to one another. The biosynthetic pathways were lead through shikimic acid and chorismic acid; the latter is a precursor of tyrosine and phenylalanine. Both amino acids are precursors of phenolic acids and 4-coumaroyl moiety which is a major substrate of flavonoid skeleton biosynthesis [22] (Appendix A). Methyl jasmonate, a biotic elicitor, was used in the elicitation experiment. Methyl jasmonate is often used as an elicitor due to the fact that it activates secondary metabolite production in plants cells as a defence response [23] (Appendix A). In addition, commercially available Plantform bioreactor temporary immersion systems (TIS), which are ideal for microshoot cultures, were used in the preliminary experiment for maintaining the *S. lateriflora* cultures at a larger biotechnological scale for the production of antioxidants (Appendix A).

## 2. Results and Discussion

### 2.1. Antioxidant Activity

In order to broadly characterise the antioxidant potential of the methanol extract obtained from the in vitro microshoot agar cultures of *S. lateriflora* (MS agar medium, supplemented with 1.0 mg/L BA and 0.5 mg/L NAA), the total phenolics and total flavonoid content, as well as the radical scavenging activity, reducing power and chelating ability, were determined; moreover, the antioxidant properties were also investigated in a biological setting.

The total phenolic content of *S. lateriflora* microshoot methanol extract was estimated using the Folin–Ciocalteu phenol reagent. In most cases, the total phenolics quantified are correlated with the antioxidant capacities of the extract, confirming the value of this assay. The total phenolic content in the extract obtained from the biomass grown in vitro was 37.270 ± 0.650 mg gallic acid equivalents (GAE)/g extract, whereas the total flavonoid content was found equal to 8.820 ± 0.900 mg quercetin equivalents (QE)/g extract (Table 1).

Flavonoids belong to the most popular antioxidants and their total amounts are very often correlated with the antioxidant potential of the extract. Antioxidants can be classified into two categories: primary (or chain-breaking) and secondary (or preventive), and the primary antioxidant reactions can also be classified as hydrogen-atom transfer (HAT) and single-electron transfer (SET). HAT occurs when an antioxidant scavenges free radicals donating hydrogen atoms, while the SET mechanism involves the transfer of a single electron to any compound causing its reduction.

In order to broadly characterise the antioxidant potential of extracts, different methods should be utilised. In this study, the primary antioxidant activity of *S. lateriflora* extract was evaluated using two methods: the DPPH (1,1-diphenyl-2-picrylhydrazyl) test (involving both HAT and SET mechanisms) and the reducing power assay (based on the SET mechanism), while the secondary antioxidant activity of the extracts was assessed by determining the ferrous ions chelating activity.

The free radical scavenging activity of *S. lateriflora* extract was determined using the DPPH test. The results showed that the extract exhibited a radical scavenging effect in a concentration-dependent manner. The radical scavenging efficiency increased with increasing concentration of the extract. At the maximum tested concentration, the activity of the extract was about 60% (Figure 1). As also indicated by IC_50_ values, the activity of the extract was found to be lower than that of butylated hydroxytoluene (BHT) used as a standard drug (IC_50_ = 1.639 ± 0.008 mg/mL and 0.066 ± 0.008 mg/mL, respectively) (Table 1).

The reducing power assay also revealed a similar trend to that confirmed in the DPPH test. *S. lateriflora* extract showed reducing power that increases with the increase in the concentration (Figure 2). The extract showed moderate activity compared to that of BHT, as confirmed by the ascorbic acid equivalent (ASE)/mL values (43.481 ± 0.237 ASE/mL and 1.131 ± 0.037 ASE/mL, respectively) (Table 1).

The results of the chelating activity assay are shown in Figure 3. The extract exhibited good, dose-dependent chelating properties, reaching approximately 80% activity at the highest tested concentrations (1 and 2 mg/mL). As also confirmed by IC_50_ values, the effect was lower than that of the reference standard EDTA (IC_50_ = 0.609 ± 0.018 mg/mL and 0.007 ± 0.001 mg/mL, respectively) (Table 1).

Based on the results obtained, it is evident that extract from *S. lateriflora* in vitro microshoot cultures acts as a weak primary antioxidant and possesses good secondary antioxidant properties. The extract contains high amounts of the flavonoids baicalin, baicalein and wogonoside, and of the phenylpropanoid glycoside, verbascoside [20]. The secondary antioxidant activity could depend mainly on high baicalein and verbascoside contents, which are considered strong iron chelators [24,25]. *Scutellaria* flavonoids have a proven strong antioxidant effect through their ability to scavenge free radicals and to chelate metal ions. These compounds also reduce the intensity of lipid peroxidation in in vitro and in vivo models. In addition to scavenging free radicals, baicalein and baicalin inhibit xanthine oxidase and protect the liposome membrane against light and H_2_O_2_-induced lipid peroxidation. Wogonin has a moderate inhibitory effect on xanthine oxidase. There are also known mechanisms of the antioxidant activity of scutellarin which, by scavenging superoxide free radicals, reduces cytotoxicity and the level of lipid peroxidation in vitro [26,27]. In vivo studies in rats have shown that baicalein, baicalin and wogonin reduce lipid peroxidation in animals when administered intraperitoneally with peroxidising agents [28].

Verbascoside has been revealed to exert antioxidant activities in many different experiments [29]. It shows H_2_O_2_ scavenging ability, superoxide anion (O_2_^−^) radical scavenging activity, nitric oxide radical scavenging activity, peroxynitrite radical scavenging activity, and DPPH radical scavenging activity. Trolox equivalent antioxidant capacity (TEAC), oxygen radical absorbance capacity (ORAC), hydroxyl radical averting capacity (HORAC), ferric reducing–antioxidant power (FRAP), and CUPRAC assay were used. Verbascoside prevents free-radical-induced hemolysis of red blood cells and protects *Saccharomyces cerevisiae* cells against superoxide anion radicals. The antioxidant activity of verbascoside is also related to its ability to inhibit malondialdehyde (MDA) generation and lipid peroxidation [30,31]. Studies of the relationships between its structure and antioxidant activities suggest that the four hydroxyls at the ortho position in the two aromatic rings of verbascoside contribute to its antioxidant activities [31].

The antioxidant efficacy of *S. lateriflora* microshoot extract was also investigated in a biological setting by evaluating the ability to protect *Escherichia coli* growth and survival from hydrogen peroxide (H_2_O_2_)-induced oxidative stress. This microbial model is a useful system to determine the antioxidant ability of phytocomplexes [32,33]. Moreover, it is simple, and cheaper compared to other cellular assays and in vitro tests.

The results showed that the extract, obtained from the in vitro cultures, exhibits a protective effect against the damage induced by H_2_O_2_ on *E. coli*. As shown in Figure 4, a 60-min growth arrest was observed in the *E. coli* cells (control group) treated with 2 mM H_2_O_2_ (Ctr+H_2_O_2_). Pretreatment with *S. lateriflora* microshoot extract (1 mg/mL) resulted in a protective effect on the bacteria, which was found to be significant starting from 20 min of exposure, in comparison to the Ctr + H_2_O_2_ group treated with the extract from *S. lateriflora* in vitro cultures (*p* < 0.01).

Figure 5 shows the results of the survival study. In the control group exposed to 10 mM H_2_O_2_ for 30 min (Ctr+ H_2_O_2_), a high reduction in viability was observed compared to the untreated Ctr group. Pretreatment with *S. lateriflora* in vitro culture extract (1 mg/mL) resulted in a protective effect, which was statistically significant in comparison to the Ctr + H_2_O_2_ group (*p* < 0.0001).

Previous studies reported that the protective effect of some plant extracts against H_2_O_2_-induced damage in *E. coli* was found to be related to their phenolic content; additionally, it was demonstrated that some pure phenolics showed remarkable protective effects on the growth and survival of *E. coli* under peroxide stress [32,34,35]. In a study carried out by Smirnova et al. [34] on some phenolics, it was hypothesised that the protective effect of the tested compounds was due to their chelating properties; in particular, the ability of polyphenols to chelate intracellular iron could contribute to the protection of *E. coli* from oxidative damage induced by H_2_O_2_. Thus, the phenolic compounds contained in the extract, obtained from in vitro microshoot cultures of *S. lateriflora*, could be mainly responsible for the observed activity, and the protective effect on *E. coli* could be related to the chelating properties of the extract.

### 2.2. Artemia salina Lethality Bioassay

The brine shrimp (*A. salina*) is an invertebrate commonly utilised for the preliminary assessment of the toxicity of bioactive compounds and plant extracts. This model is most suitable for evaluating toxicity as it is cost-effective, easy to use, and allows rapid testing. The brine shrimp lethality bioassay may be considered an alternative to assays based on in vitro cell culture [36] and in vivo assays. It has been reported that the results of oral acute toxicity determination in the murine model correlate well with that of the *A. salina* lethality bioassay. Thus, this bioassay is a useful tool for predicting the acute toxicity of plant extracts [37].

Aimed at establishing the potential safety of the *S. lateriflora* microshoot extract, an *A. salina* lethality bioassay was performed. The extract was found to be non-toxic against brine shrimps; in fact, the median lethal concentration (LC_50_) calculated for the extract resulted in >1000 μg/mL.

The revealed antioxidant potential of *S. lateriflora* microshoot extracts and the results obtained after the optimization of agar and agitated cultures [20,21,38] encouraged us to assess whether metabolite production and accumulation in biomass could be improved using different biotechnological strategies.

### 2.3. Enhanncing the Production of Secondary Metabolites

#### 2.3.1. Agar Culture Feeding with Biosynthetic Precursors

The microshoot cultures of *S. lateriflora* were maintained on LS medium containing 1.0 mg/L BA and 0.5 mg/L NAA, which was recognised based on our previous study as the best production medium for this type of culture. Phenylalanine and tyrosine were added to the culture medium as biosynthetic precursors of phenolic compounds at a concentration of 1 g/L when the culture was established. Throughout the cultivation period (4 weeks), the cultures were in contact with the medium containing the selected amino acid. The phenylalanine-supplemented cultures showed no significant effect on biomass growth, whereas the biomass growth in cultures supplemented with tyrosine was, on average, 1.28-fold weaker compared to the control cultures.

Extracts from the biomass obtained from precursor-treated cultures contained the same compounds as the biomass obtained from the control cultures (i.e., baicalein, baicalin, wogonin, wogonoside, scutellarin, oroxylin A, and verbascoside) (Figure 6). The total flavonoid content in the biomass obtained from phenylalanine-supplemented cultures was estimated at 2531 mg/100 g DW, and in the biomass obtained from tyrosine-supplemented cultures at 1263 mg/100 g DW (Table 2). These values were lower than the flavonoid content determined in the extracts obtained from the control cultures (2768 mg/100 g DW). Among the flavonoids, the dominant compound in control and experimental cultures was baicalin (997–1692 mg/100 g DW). After phenylalanine administration, the content of wogonin and scutellarin was found to be higher, the content of baicalein and baicalin was comparable, and the content of wogonoside and oroxylin A was lower than in the control cultures. Although these results are satisfactory, it does not support the use of phenylalanine as a precursor in the enhanced biosynthesis of these metabolites. In the extracts obtained from tyrosine-treated cultures, the content of scutellarin was 16.5-fold higher compared to the control cultures, while the content of other flavonoid metabolites was lower.

The content of verbascoside in the biomass obtained from phenylalanine-supplemented cultures was 458 mg/100 g DW, and in the biomass from tyrosine-supplemented cultures was 554 mg/100 g DW, which were 1.57- and 1.89-fold higher than those of the control cultures, respectively (Table 2). These results support the utilization of both amino acids in the enhanced biosynthesis of verbascoside.

Although a high content of secondary metabolites was found in the biomass, the agar cultures of *S. lateriflora* are not the best model for obtaining bioactive compounds. This is due to the fact that these cultures are characterised by poor (1.67–1,93-fold) biomass growth, which means that the amount of the obtained biomass, despite high metabolite content, will also be lower. Lower flavonoid accumulation in biomass was found after the administration of phenylalanine in comparison with the control, while tyrosine administration induced a more than two-fold decrease in the total flavonoid content in the cultures. These results differ from those confirmed by us in the cultures of other skullcap species. For instance, in the cultures of *S. baicalensis* maintained in our laboratory, the same precursors stimulated the production of flavonoids, and tyrosine was found to be more conducive to flavonoid accumulation than phenylalanine [22]. In *Mentha longifolia* agar callus and shootlet cultures, phenylalanine administration (15 mg/L) resulted in an increase in the production of rosmarinic acid. The authors obtained increased levels of total phenols as well [39]. These results mean that each plant tissue and type of culture require a proper concentration of phenylalanine to stimulate phenylalanine ammonia-lyase (PAL) activity, which is the key enzyme in the phenolic biosynthesis pathway. An increased content of verbascoside was found in tyrosine-supplemented cultures. These results are in line with those documented in *S. lateriflora* cultures in the present study.

#### 2.3.2. Agitated Cultures

As our previous study revealed higher biomass growth rates in agitated cultures than in agar cultures and a better uptake of components from a liquid medium, a similar but broader feeding experiment with agitated cultures was planned for the present study. Based on the previous results, agitated cultures of *S. lateriflora* were maintained on LS medium containing 1.0 mg/L BA and 1.0 mg/L NAA, as it was identified as the best production medium.

##### Feeding with Biosynthetic Precursors

Phenylalanine and tyrosine were administered to the medium at four different concentrations (1.0, 1.5, 2.0, and 2.5 g/L). Since higher concentrations were chosen than in agar cultures, the amino acids were administered into the media after 3 weeks of culture growth. In addition, two different exposure periods (3 and 7 days) were considered. The biomass increments of cultures supplemented with phenylalanine ranged from 6.0- to 7.2-fold (day 3) and from 6.7- to 8.1-fold (day 7) after precursor administration. These results were comparable to, or even better than, those documented in the control cultures (6.0 and 6.5-fold, respectively), which indicated that phenylalanine did not inhibit the growth of the cultures at the tested concentrations. On the other hand, cultures supplemented with tyrosine showed only a mild increase in growth from 2.8- to 3.6-fold (day 3) and 3.2- to 4.6-fold (day 7) after precursor administration. The biomass of *S. lateriflora* cultures darkened due to the addition of tyrosine, especially with higher concentrations. Analyses of the effect of phenylalanine (0.016–1.6 g/L) on the growth of *Aronia* ssp. showed that the precursor added to the cultures stimulated the growth of *Aronia melanocarpa* and *Aronia arbutifolia* at a concentration range of 0.016–0.16 and 0.016–0.8 g/L, respectively. At higher concentrations of precursor, in contrast to *S. lateriflora*, the growth of *Aronia* ssp. cultures was inhibited [40].

Biomass extracts of *S. lateriflora* obtained from the cultures supplemented with both phenylalanine and tyrosine and from the control cultures were found to contain the following flavonoids: baicalein, baicalin, wogonin, wogonoside, scutellarin, oroxylin A, and verbascoside. The total flavonoid content determined in the phenylalanine-supplemented cultures ranged from 2414 to 3042 mg/100 g DW (day 3) and from 2964 to 3765 mg/100 g DW (day 7) after precursor administration. In contrast, the total flavonoid content in the tyrosine-supplemented cultures ranged from 1176 to 1932 mg/100 g DW (day 3) and from 713 to 1799 mg/100 g DW (day 7) after precursor administration (Appendix A). All of the values determined after phenylalanine administration were significantly higher (1.39–1.75-fold on day 3 and 1.76–2.24-fold on day 7, respectively) compared to the values determined in the control culture (1740 mg/100 g DW on day 3 and 1680 mg/100 g DW on day 7, respectively). In the case of tyrosine administration, the total flavonoid content determined in the cultures was slightly higher (1.07–1.11-fold) than in the control, but only by low concentrations of the precursor (i.e., 1.0 and 1.5 g/L) (Figure 7 and Figure 8). The administration of higher doses of tyrosine caused a drastic decrease in flavonoid content.

The main flavonoid found in all of the biomass extracts of phenylalanine-supplemented cultures was wogonoside (maximum: 1149 mg/100 g DW). In addition, the content of wogonin and baicalin was very high (maximum: 965 and 988 mg/100 g DW, respectively) (Appendix A). In the biomass extracts of tyrosine-supplemented cultures, wogonin was found to be dominant on day 3 (maximum: 747 mg/100 g DW), while on day 7 wogonoside was also the main metabolite (maximum: 655.5 and 517 mg/100 g DW, respectively) (Appendix A). The maximum content of almost all individual flavonoids determined after both phenylalanine and tyrosine administration on days 3 and 7 was higher compared to the control samples. Only on day 3 after tyrosine administration was comparable content to the control samples estimated for baicalin and wogonoside. The highest content of flavonoids was obtained with lower tyrosine concentrations of 1.0 and 1.5 g/L, and with average applied phenylalanine concentrations of 1.5 g/L (day 3) and 2.0 g/L (day 7) (Table 3 and Table 4).

The verbascoside content of the extracts ranged from 327.7 to 474.8 mg/100 g DW after phenylalanine administration and from 128.3 to 299.5 mg/100 g DW after tyrosine administration (Appendix A), with the highest content confirmed for lower precursor concentrations (1.0 and 1.5 g/L). In the extracts obtained from the biomass harvested on day 3, the content of this compound was higher than in the control for all of the tested phenylalanine concentrations (Appendix A). The maximum content of verbascoside, determined after phenylalanine administration was 1.5–1.78-fold higher compared to the control cultures (311 and 267 mg/100 g DW) (Table 3 and Table 4). This indicates that the aromatic amino acids added to the liquid medium are taken up and incorporated into the biosynthesis pathways of phenolic metabolites.

The biosynthesis pathways of compounds having a phenolic structure are related due to the similarities in their chemical structure (Appendix A). Phenylalanine is the main precursor of all phenolic acids. Similar to tyrosine, it is formed from shikimic and chorismic acids [30,41]. The key enzyme of tyrosine metabolism, tyrosine ammonia-lyase (TAL), is also less studied than PAL. Both tyrosine and phenylalanine are also precursors of flavonoids. The biosynthesis of flavonoids is associated with two metabolic pathways: the malonic acid pathway and the shikimic acid pathway. The key enzymes involved in the further steps of flavonoid biosynthesis are chalcone synthase and chalcone isomerase [42]. The central pathway of flavonoid biosynthesis in plants is quite conservative. Nevertheless, depending on the species, a group of enzymes modify the basic flavonoid skeleton, leading to the different flavonoid subclasses. The administration of precursors upstream of the pathway is justified if the biosynthesis and accumulation of the whole group of metabolites is expected to increase. The results of the present study showed that phenylalanine supplementation of the medium in agitated cultures was beneficial and an increase in total flavonoid and verbascoside content was noted for all of the tested precursor concentrations. Moreover, phenylalanine did not induce growth inhibition of the biomass. In contrast, metabolite accumulation in the biomass of tyrosine-supplemented cultures was much lower. Based on the findings, it can be concluded that supplementation with only the lowest concentration of tyrosine can yield beneficial results. Furthermore, tyrosine administration had a negative effect on the growth and appearance of biomass. These results differ from those documented in our *S. baicalensis* cultures, in which the administration of tyrosine resulted in higher flavonoid and verbascoside content compared to phenylalanine [22].

Phenylalanine is commonly administered in in vitro cultures as a precursor of phenolic compounds. Numerous scientific studies have focused on analyzing metabolite content after administration of this amino acid, and some have also investigated the activation of particular enzymes under the influence of this precursor. Tyrosine supplementation is not a popular procedure in plant biotechnology. It is also less convenient to use due to its lower solubility in aqueous solutions. Earlier studies have demonstrated an increase in the accumulation of phenolic compounds such as flavonoids, anthocyanidins, or phenolic acids after phenylalanine administration in cell cultures of different plant species. Phenylalanine concentration 0.75 g/L highly stimulated quercetin production in a callus culture of *Abutilon indicum* [43]. In cell cultures of *Ginkgo biloba* increased levels of gallic, protocatechuic and p-hydroxybenzoic acids were detected after administration of 0.66–1.33 g/L Phe [44]. Verbascoside biosynthesis, in contrast, was activated in cell suspension cultures of *Buddleja cordata* by the addition of 100 mg/L of Phe [45]. However, a much lower concentration of phenylalanine (5–10 mg/L) resulted in anthocyanin accumulation in the cell suspension culture of *Panax sikkimensis* [46].

Also, in shoot cultures of species such as *Exacum affine* [47], *Nasturtium officinale* [48], *Aronia melanocarpa* and *A. arbutifolia* [40], the influence of phenylalanine administration on the biosynthesis of phenolic metabolites was investigated. It has been shown that the optimum phenylalanine concentration that can stimulate the production of phenolic acids in shoot cultures varies widely (e.g., 0.016 g/L for *A. melanocarpa*, 0.16 g/L for *A. arbutifolia*, 0.5 g/L for *N. officinale* and 1.6 g/L for *E. affine*). The demonstrated differences in the optimal concentration of phenylalanine may be caused by both the plant species and the type of the tested culture. Phe can be used in the cells as a substrate for metabolite biosynthesis (higher concentrations) or act as an activator of phenylalanine ammonia-lyase (lower concentrations). The method and scheme of the precursor addition are also crucial; for example, Edahiro et al. proved that the repeated administration of phenylalanine to a suspension culture of strawberry cells (*Fragaria* × *ananassa*) resulted in the continuous production of anthocyanins [49].

##### Elicitation with Methyl Jasmonate

Agitated cultures were grown on the same variant of LS medium (PGRs) and following the same experiment scheme as for precursor administration. Methyl jasmonate was used as an elicitor at concentrations of 10, 50, and 100 µM. The biomass increments after methyl jasmonate administration ranged from 2.0- to 4.8-fold (day 3) and 2.4- to 5.9-fold (day 7). Administration of the highest concentration of methyl jasmonate resulted in growth inhibition of the biomass. Likewise, the best growth in cell cultures of different *Ocimum* species [50] was documented after induction with 25–50 µM methyl jasmonate.

Baicalein, baicalin, wogonin, wogonoside, scutellarin, oroxylin A, and verbascoside were the metabolites detected in the biomass extracts of cultures elicited both for 3 and 7 days. The same metabolites were also found in the control cultures. The maximum content of individual flavonoids (baicalein, baicalin, scutellarin, wogonin, and oroxylin A) was higher in the elicited cultures (Table 3 and Table 4). The dominant compound in the cultures was wogonin. The total flavonoid content determined after methyl jasmonate administration ranged from 976.6 to 2059.5 mg/100 g DW after 3 days (Appendix A) and from 762.3 to 1814.9 mg/100 g DW after 7 days of culture (Appendix A). On both days, when lower concentrations of methyl jasmonate (10 and 50 µM) were administered, higher total flavonoid content was estimated in the elicited cultures compared to the control cultures (1740 mg/100 g DW and 1680 mg/100 g DW) (Figure 7 and Figure 8).

The verbascoside content of the extracts varied widely in the tested cultures and ranged from 20.7 to 381.5 mg/100 g DW on day 3 (Appendix A) and from 41.9 to 121.5 mg/100 g DW on day 7 (Appendix A). The highest content of this compound was noted on day 3 with an elicitor concentration of 10 µM (Table 3). Only with this elicitor concentration was the content of verbascoside found to be higher than in the control cultures (311 mg/100 g DW and 267 mg/100 g DW).

The mechanisms of elicitation are much more complex than those of the stimulation of metabolite biosynthesis by precursor administration [51]. Many authors have described that the addition of biotic and abiotic elicitors caused an increase in the synthesis of secondary metabolites in in vitro cultures. Methyl jasmonate as a PGR is a biotic elicitor [23]. In plants, methyl jasmonate induces a defence response to injury or pathogen attack by activating a number of enzymes involved in the biosynthesis of polyphenolic metabolites. These include phenylalanine ammonia-lyase, stilbene synthase, anthocyanidin synthase, chalcone isomerase, and also catalase [23]. Of these, phenylalanine ammonia-lyase is particularly important because it plays a key role in the synthesis of metabolites from phenylalanine [35] (Appendix A). Methyl jasmonate acts by receptor, and through signal transduction cascade, generates reactive oxygen species. As a result, the biosynthesis of secondary metabolites increases in cells. The intracellular signaling pathway involves a cascade of radical reactions associated with the expression of genes encoding enzymes and other proteins responsible for protective mechanisms. The elicitation effect of an elicitor is determined by its concentration, the exposure time of the cultures, and the growth stage (age) of the in vitro cultures at the time of elicitation. Several studies have demonstrated the effect of methyl jasmonate on the biosynthesis of phenolic compounds or the increase in their content under the influence of this elicitor in in vitro cultures of different plant species. Suspension cultures are the most frequently used research model. In cell cultures of *Buddleja cordata*, 50 µM of methyl jasmonate increases the verbascoside level 213% above the control [45]. Rosmarinic acid biosynthesis was increased in *Satureja khuzistanica* cultures by administering jasmonate at a concentration of 50 mg/L [52], and in cultures of *Mentha piperita* [53] at a concentration of 50–200 µM. Other authors demonstrated, in *Thevetia peruviana* cultures, [54] that exposure to 3 μM MeJA increased phenolic content 1.49-fold, and flavonoid content 2.55-fold, compared to the control culture. The addition of 10 µM methyl jasmonate resulted in a positive effect on the production of anthocyanins and catechins in petiole cell cultures of *Vitis vinifera* [55]. The effect of different concentrations of methyl jasmonate on the production of secondary metabolites in plant organ cultures was also investigated. Methyl jasmonate (0.6, 1.2, 2.5 mg/L) in root cultures of *Ajuga bracteosa* elevated total phenolic content and total flavonoid content depending on the culture growth phase [56]. Elicited shoot and root cultures of *Hypericum perforatum* produced flavonoids and naphthodianthrones [57], while 100 µM methyl jasmonate was the most effective for flavonoid production in shoot cultures of *Nasturtium officinale* [58] and phenolic acids in shoot cultures of *Exacum affine* [48]. In the agitated shoot cultures of *Schisandra chinensis*, methyl jasmonate either moderately decreased or did not affect the accumulation of dibenzocyclooctadiene lignans (depending on the concentration used) [59]. In the present work, the total flavonoid content in *S. lateriflora* cultures was higher than that in the control cultures, and significantly higher than the content determined in our *S. baicalensis* cultures (max. 310 mg/100 g DW) [22]. However, it should be emphasised that with the highest concentration of methyl jasmonate (100 µM), the metabolite content of *S. lateriflora* microshoots decreased twofold.

##### Combined Strategy: Feeding with Biosynthetic Precursors and Elicitation

We decided to combine both feeding and elicitation to increase metabolite accumulation by administering a precursor and an elicitor simultaneously to the culture medium. Agitated cultures of *S. lateriflora* were maintained in an identical LS medium as used in previous experiments, with the same final concentrations of biosynthetic precursors (1.0, 1.5, 2.0, and 2.5 g/L). However, based on our previous experiments on *S. baicalensis* [22] and *Nasturtium officinale* [58], a concentration of 50 µM was chosen for the elicitor. The biomass increments of cultures in which phenylalanine and methyl jasmonate were administered ranged from 5.8- to 7.1-fold (day 3) and from 6.4- to 8.2-fold (day 7) after precursor administration. When tyrosine and methyl jasmonate were administered, the biomass increments ranged from 2.9- to 3.8-fold (day 3) and from 3.0- to 4.3-fold (day 7). After the administration of tyrosine with methyl jasmonate, biomass darkening was found for all of the tested concentrations of the precursor. The results showed that simultaneous administration of the precursor and elicitor did not significantly affect biomass compared to the administration of precursors alone.

Baicalein, baicalin, wogonin, wogonoside, scutellarin, oroxylin A, and verbascoside were found in the biomass extracts of cultures supplemented with phenylalanine and elicitor, as well as that of the cultures containing tyrosine and elicitor. An identical qualitative composition was noted in the biomass extracts of the control cultures and those of cultures treated with precursors or methyl jasmonate alone. The maximum total flavonoid content determined after the administration of phenylalanine with the elicitor was 2526 mg/100 g DW (day 3) and 2617 mg/100 g DW (day 7), while the content determined after the administration of tyrosine with the elicitor was 1906 mg/100 g DW (day 3) and 1799 mg/100 g DW (day 7). Lower total flavonoid content was obtained in the cultures in which the precursor was administered together with the elicitor, in comparison to the cultures administered with phenylalanine or tyrosine alone (Figure 7 and Figure 8). The total flavonoid content determined after the administration of methyl jasmonate with phenylalanine at all of the tested concentrations was higher compared to the control samples (both on days 3 and 7). However, the total flavonoid content determined after the administration of tyrosine together with methyl jasmonate was higher than the control values only for the amino acid concentration of 1 g/L.

The main flavonoid detected in all of the extracts after the administration of phenylalanine with the elicitor was wogonin (maximum content: 783 mg/100 g DW). Baicalin and wogonoside were also detected in high amounts (maximum content: 630 and 620 mg/100 g DW, respectively). Furthermore, after the administration of tyrosine with the elicitor, wogonin, wogonoside, and baicalin were quantitatively dominant in the extracts (maximum content: 659, 479, and 507 mg/10 0g DW, respectively) (Appendix A). The maximum content of individual flavonoids determined after the administration of phenylalanine with elicitor was higher compared to their content in the control samples on both days 3 and 7. The highest metabolite contents were confirmed at a phenylalanine concentration of 1.5 g/L. After the administration of tyrosine with methyl jasmonate, the maximum content of all of the individual metabolites (except wogonin) was found to be higher than those of the control samples. However, higher metabolite contents were determined on both days 3 and 7 only for the lower tyrosine concentrations (Appendix A).

The verbascoside content of the extracts also varied widely. The maximum content obtained after phenylalanine administration with methyl jasmonate was 160.6 mg/100 g DW on day 3 and 170.7 mg/100 g DW on day 7. Compared to the control cultures, these values were 1.94- and 1.56-fold lower, respectively (Table 3 and Table 4). After the administration of tyrosine with methyl jasmonate, the content of verbascoside decreased 6.06–12.22-fold compared to the control cultures (25.46 and 44.01 mg/100 g DW, respectively) (Table 3 and Table 4).

Theoretically, the combined strategy should benefit both elicitation and precursor feeding. However, the results obtained in the study indicate that too many stimuli, and activation of multiple signaling pathways, did not have any positive effect on metabolite production in *S. lateriflora* cultures. When methyl jasmonate was combined with phenylalanine or tyrosine, the obtained amounts of metabolites were lower compared to those documented with the administration of the precursors alone. This was evident in the content of both flavonoids and verbascoside. On the other hand, in the case of our *S. baicalensis* shoot cultures, with the administration of both precursors in combination with the same elicitor, the flavonoid content determined on day 3 was higher than the content confirmed after the administration of the precursors alone, and on day 7 was comparable [22]. The content of verbascoside was higher in *S. baicalensis* cultures administered with phenylalanine or tyrosine, and methyl jasmonate combinations compared to those administered with precursors alone. In contrast to *S. lateriflora*, in the biomass of *S. baicalensis* cultures, significantly better results were confirmed after the administration of tyrosine with the elicitor [22]. Summarised, different results were obtained in *S. lateriflora* and *S. baicalensis* in vitro cultures, despite the fact that these species are taxonomically very closely related (i.e., belong to the same genus). The parallel administration of Phe and methyl jasmonate into *Exacum affine* culture medium resulted in higher contents of chlorogenic and caffeic acids in comparison with a single treatment (precursor or elicitor). This worked when 100 µM of methyl jasmonate was used. With a higher elicitor concentration (800 µM), the obtained effect was the opposite [48]. It indicated that the influence of feeding and elicitation should be investigated experimentally for each in vitro culture separately. This may mean that the effect of a given strategy is not influenced by species specificity or complexity of the biosynthetic pathway of the tested metabolites, but by the sensitivity of the in vitro culture to the factors added to the medium.

#### 2.3.3. Cultures in Bioreactors

The microshoot cultures of *S. lateriflora* were also maintained for the same time in a commercially available Plantform temporary immersion system bioreactor (TIS) on two different media containing 1.0 mg/L BA and 0.5 mg/L NAA for 4 weeks. We selected the MS and LS medium variants based on our previous study which showed that these media were the most conducive to flavonoid accumulation in biomass. The increase in the dry biomass content in bioreactor cultures was satisfactory and comparable to that achieved in agitated cultures (6.3-fold on MS medium and 6.9-fold on LS medium).

Baicalin, wogonin, wogonoside, scutellarin, oroxylin A and verbascoside were found in all of the extracts obtained from biomass grown in the Plantform bioreactors. However, baicalein was not detected.

The total flavonoid content in the extracts obtained from the biomass grown in the Plantform bioreactors on MS and LS medium was 1783 and 2348 mg/100 g DW, respectively (Table 5). This was 2.55-fold and 2.74-fold higher compared to the content obtained for the same MS medium variant in agar and agitated cultures, respectively. For the LS medium variant, the total flavonoid content was 1.18-fold lower than in agar cultures and 1.74-fold higher than in agitated cultures [20]. Baicalin was the main compound in the bioreactor cultures. The content of verbascoside was found to be higher in cultures grown on MS medium (485.85 mg/100 g DW) than on LS medium (310.37 mg/100 g DW) (Table 5).

The in vitro shoot cultures of *S. lateriflora* did not show good biomass increments in any culture type. However, the biomass increments obtained in bioreactors, as well as the flavonoids and verbascoside contents, were very promising. The total flavonoid content in bioreactor cultures was 5.6-7.4-fold higher than in the *S. baicalensis* cultures maintained in the same type of bioreactor. On the other hand, verbascoside production was 1.8-1.9-fold lower than in Baikal skullcap cultures [22].

The optimization of cultivation depends on the type of bioreactor used and its operation, including aeration and immersion time, which is widely reported in the literature [60,61]. This has been demonstrated by us for both *Schisandra chinensis* [62] and *Verbena officinalis* cultures [63]. It was shown that the operation conditions influenced the accumulation of flavonoids, phenolic acids, and phenylpropanoid glycosides. The same groups of phenolic metabolites were also detected in the presented experiment on *S. lateriflora*. The next step of our experiment should be the optimization of *S. lateriflora* culture conditions in bioreactors. After the optimization of the metabolite production process, the bioreactor cultures of *S. lateriflora* could be applied to produce valuable products on a commercial level.

## 3. Materials and Methods

### 3.1. In Vitro Cultures

In vitro *S. lateriflora* microshoot cultures were derived from 4-week-old seedlings germinated on the Murashige and Skoog (MS) [64] initiating medium. The medium contained 3% (*w*/*v*) sucrose, 0.7% (*w*/*v*) phytoagar (Duchefa Biochemie, Haarlem, The Netherlands) as a gelling agent, and 2.0 mg/L 6-benzyladenine (BA) and 2.0 mg/L 1-naphthaleneacetic acid (NAA). The pH of the medium was adjusted to 5.7 before autoclaving. The seeds used for germination were obtained in 2015 from W.J. Beal Botanical Garden, Michigan State University (USA). The cultures were maintained on the MS medium supplemented with 1.0 mg/L BA and 0.5 mg/L NAA [20].

#### 3.1.1. Agar Cultures

Experimental *S. lateriflora* agar cultures were maintained on Linsmaier and Skoog (LS) medium [65] supplemented with 1.0 mg/L BA and 0.5 mg/L NAA. We chose this medium as our previous experiments documented that it was the best “productive” medium among those tested for this type of in vitro culture [20]. Two biosynthetic precursors of phenolic compounds, namely phenylalanine and tyrosine, were added to the medium, both at a concentration of 1.0 g/L. The doses of precursors to be added were determined in our previous study with *S. baicalensis* [22]. Throughout the growth cycle (4 weeks, five series), the cultures were grown under constant (24/24 h), artificial light conditions (16 μmol/m^2^/s, LF-40 W lamp, daylight; Piła, Poland) at 25 ± 2 °C. After cultivation, the biomass was harvested and dried at 40 °C in a drying oven. The increase in biomass was calculated by dividing the sample dry weight (DW) by the dry weight of the inoculum (0.5 g of fresh biomass).

#### 3.1.2. Agitated Cultures

Based on our previous experiments, the LS medium supplemented with 1.0 mg/L BA and 1.0 mg/L NAA was chosen for agitated cultures as the best “productive” medium [20]. The cultures were maintained for 3 weeks (three series) in liquid medium (150 mL) in a 500 mL Erlenmeyer flask with 0.5 g of fresh biomass inoculum, on a rotary shaker (Altel, Łódź, Poland), which was operating at a speed of 140 rpm with an amplitude of 35 mm. The other conditions of cultivation were the same as those mentioned for agar cultures. After 3 weeks, biosynthetic precursors (phenylalanine and tyrosine) and an elicitor (methyl jasmonate) were added to the medium at the final following concentrations: phenylalanine and tyrosine—1.0, 1.5, 2.0, and 2.5 g/L; and methyl jasmonate—10, 50, and 100 µM, respectively. The doses of the precursors and elicitor to be added were determined in our previous study with *S. baicalensis* [22]. In the present study, we decided to administer the precursor and elicitor together and added 50 μM methyl jasmonate and one of the precursors (phenylalanine or tyrosine) at the final concentrations of 1.0, 1.5, 2.0, and 2.5 g/L to the medium. All supplemented experimental and control cultures were grown for the next 3 or 7 days. After cultivation, the biomass was harvested and dried at 40 °C in a drying oven. The increase in biomass was calculated by dividing the sample DW by the DW of the inoculum.

#### 3.1.3. Cultures in Bioreactors

The biosynthetic potential of *S. lateriflora* microshoot cultures was also investigated using the bioreactor culture model. Based on our previous results, we chose MS and LS media which were identified as the best “productive” media [20]. The temporary immersion system (TIS) Plantform bioreactors (PlantForm, Hjärup, Sweden) were inoculated with 9 g of microshoots per 500 mL of medium (MS or LS) supplemented with 1.0 mg/L BA and 0.5 mg/L NAA. The immersion frequency was 5 min every 90 min. Cultures were grown for 4 weeks, in light and temperature conditions as described for agar cultures. After cultivation, the biomass was harvested and dried at 40 °C in a drying oven.

### 3.2. Reverse-Phase High-Performance Liquid Chromatography (RP-HPLC) Analysis

The dried and pulverised biomass samples were extracted with 50 mL of analytical-grade methanol for 2 h at 78 ± 2 °C under a reflux condenser. After extraction, the obtained extracts were filtered through a paper filter, transferred to crystallisers, and left at room temperature to allow the solvent to evaporate. The remains were dissolved in 4 mL of HPLC-grade methanol (Merck) and subjected to HPLC. Qualitative and quantitative HPLC with diode array detection (DAD) was performed as described previously [20,66]. Compounds were identified based on their retention times and UV spectra, as well as by comparison with reference substances (27 flavonoids, 19 phenolic acids, benzoic and cinnamic acids, 2 phenylethanoid glycosides). The reference standards used for the HPLC analyses of skullcap-specific flavonoids were baicalein, baicalin, scutellarin, wogonin (ChromaDex, Irvine, CA, USA), chrysin, scutellarein, wogonoside (Sigma–Aldrich Co., Saint Louis, MO, USA), oroxylin A and skullcapflavone II (ChemFaces Biochemical Co., Wuhan, China). The verbascoside was purchased form ChromaDex, Irvine, CA, USA.

### 3.3. Total Phenolic and Flavonoid Content

The total phenolic content of the methanol extract of in vitro *S. lateriflora* culture maintained on the agar MS medium supplemented with 1.0 mg/L BA and 0.5 mg/L NAA was measured using the Folin–Ciocalteu reagent as previously reported [61]. Briefly, 100 μL of a solution containing MeOH extract at an appropriate concentration was mixed with 200 μL of Folin–Ciocalteu reagent, 2 mL of distilled water, and 1 mL of 15% sodium carbonate. The resulting mixture was incubated at room temperature in the dark for 2 h. Then, the absorbance of the solution was measured using a spectrophotometer at a wavelength of 765 nm. Gallic acid was used as a standard, and the content of total phenolics was expressed as mg of gallic acid equivalents (GAE)/g extract (DW) ± standard deviation (SD).

The total flavonoid content of the extract was determined using the aluminium chloride colorimetric assay [67]. Briefly, 500 μL of each appropriately diluted sample solution was mixed with 1.5 mL of MeOH, 100 μL of 10% aluminium chloride, 100 μL of 1 M potassium acetate, and 2.8 mL of distilled water. The reaction mixtures were incubated at room temperature in the dark for 30 min, and their absorbance was measured at a wavelength of 415 nm. The calibration curve was generated using quercetin as a standard, and the total flavonoid content was expressed as mg of quercetin equivalents (QE)/g extract (DW) ± SD.

### 3.4. Antioxidant Activity

Since antioxidant activity is associated with different mechanisms, a single testing method cannot provide a comprehensive view of the antioxidant profile of a sample. Therefore, the antioxidant capacity of plant-derived phytocomplexes or isolated compounds is evaluated using various methods. In this study, the in vitro antioxidant activity of methanol extract from the in vitro *S. lateriflora* microshoots maintained on the agar MS medium supplemented with 1.0 mg/L BA and 0.5 mg/L NAA was assessed using three in vitro tests that are based on different mechanisms: the DPPH assay, the reducing power assay, and the ferrous ion chelating assay. Additionally, the antioxidant efficacy of *S. lateriflora* extracts was investigated in a biological setting using *E. coli* by evaluating the ability of the extracts to protect bacterial growth and survival against H_2_O_2_-induced oxidative stress.

#### 3.4.1. Free Radical Scavenging Activity

The free radical scavenging activity of *S. lateriflora* extract was determined using the DPPH method [68]. The extract was tested at different concentrations ranging from 0.0625 to 2 mg/mL. An aliquot (0.5 mL) of a methanol solution containing different concentrations of the extract was added to 3 mL of freshly prepared methanol–DPPH solution (0.1 mM). After 20 min of initial mixing, the change in the optical density of the solution was measured at a wavelength of 517 nm using a model UV-1601 spectrophotometer (Shimadzu). Butylated hydroxytoluene (BHT) was used as a reference. The scavenging activity of the sample solution was measured as a decrease in its absorbance in comparison to the DPPH standard solution. The results averaged from three independent experiments were expressed both as mean radical scavenging activity (%) ± SD and mean 50% inhibitory concentration (IC_50_) ± SD.

#### 3.4.2. Reducing-Power Assay

The reducing power of *S. lateriflora* extract was evaluated by spectrophotometric detection of Fe^3+^-Fe^2+^ transformation method [69]. The extract was tested at different concentrations ranging from 0.0625 to 2 mg/mL. Solutions of different concentrations of extract in 1 mL solvent were mixed with 2.5 mL of phosphate buffer (0.2 M, pH 6.6) and 2.5 mL of 1% potassium ferricyanide [K_3_Fe(CN)_6_], and the resulting mixture was incubated at 50 °C for 20 min. The solution was cooled rapidly, mixed with 2.5 mL of 10% trichloroacetic acid, and centrifuged at 3000× *g* rpm for 10 min. After centrifugation, the supernatant (2.5 mL) was mixed with 2.5 mL of distilled water and 0.5 mL of 0.1% fresh ferric chloride (FeCl_3_). The absorbance of the solution was measured at a wavelength of 700 nm after 10 min. The increase in the absorbance of the reaction mixture indicates an increase in its reducing power. An equal volume (1 mL) of water mixed with a solution prepared as described above was used as a blank. Ascorbic acid and BHT were used as references. The results averaged from three independent experiments were expressed as mean absorbance values ± SD. The reducing power was also expressed as ascorbic acid equivalent (ASE/mL); when the reducing power is 1 ASE/mL, the reducing power of 1 mL extract is equivalent to 1 μmol ascorbic acid.

#### 3.4.3. Ferrous Ions (Fe^2+^) Chelating Activity

The Fe^2+^ chelating activity of *S. lateriflora* extract was determined based on the formation of Fe^2+^–ferrozine complex [70]. The extracts were tested at different concentrations ranging from 0.0625 to 2 mg/mL. Briefly, an extract of different concentrations in 1 mL of solvent was mixed with 0.5 mL of methanol and 0.05 mL of 2 mM FeCl_2_. Then, the reaction was initiated by adding 0.1 mL of 5 mM ferrozine. The reaction mixture was shaken vigorously and left at room temperature for 10 min. After incubation, the absorbance of the solution was measured spectrophotometrically at a wavelength of 562 nm. A solution containing FeCl_2_ and ferrozine complex formation molecules was used as control. Ethylenediaminetetraacetic acid (EDTA) was used as a reference. The results averaged from three independent experiments were expressed as mean inhibition of the ferrozine-(Fe^2+^) complex formation (%) ± SD and IC_50_ ± SD.

#### 3.4.4. Protective Effect on *Escherichia coli* Growth and Survival under Peroxide Stress

The ability of *S. lateriflora* extract to protect bacterial growth and survival against H_2_O_2_-induced oxidative stress was evaluated using a protocol described previously [71]. *Escherichia coli* ATCC 25922, obtained from the Department of Scienze Chimiche Biologiche Farmaceutiche ed Ambientali, University of Messina (Messina, Italy), in-house culture collection, was cultured in LB medium. The bacterial suspension (OD600  =  0.2) was pretreated with the extracts (1 mg/mL) before the analysis. To determine the ability of *Scutellaria* extract to protect *E. coli* cells against oxidative stress-induced growth inhibition, the bacteria (OD600  =  0.4) was treated with H_2_O_2_ (2 mM) and the growth was monitored for 3 h.

For survival studies, the bacterial suspension (OD600  =  0.4) was exposed to a higher concentration of H_2_O_2_ (10 mM), which caused a bactericidal effect. After 24 h, cell survival was estimated by counting the number of viable colonies. Quercetin (0.2 mM) was used as a reference standard. The results averaged from three independent experiments were expressed as mean absorbance ± SD and survival (%)  ±  SD for the protective effect of extracts on *E. coli* growth and survival, respectively.

### 3.5. Brine Shrimp (Artemia salina) Lethality Bioassay

To investigate the potential toxicity of *S. lateriflora* extract, the median lethal concentration (LC_50_) was determined as described previously [72]. Briefly, the extract was dissolved and diluted in artificial seawater, and then transferred to vials at a final concentration of 10, 100, 500, and 1000 µg/mL. Ten brine shrimp larvae were transferred to each sample vial, followed by which artificial seawater was added to obtain a final volume of 5 mL. After 24 h of incubation (25–28 °C), the number of surviving larvae were counted in each vial. The assay was carried out in triplicate, and the LC_50_ values were determined using the Litchfield and Wilcoxon method. Extracts are considered non-toxic if the LC_50_ is above 1000 µg/mL.

### 3.6. Statistical Analysis

All experiments were performed with at least three independent replications. For the antioxidant bioassays, the statistical comparisons of the data were performed using a Student’s t-test for unpaired data. *p*-values of < 0.05 were considered statistically significant. The statistical analysis of metabolite content was conducted using the STATISTICA 13.3 software program (TIBCO Software Co., Palo Alto, CA, USA). The level of significance was set at *p* < 0.05. The differences in results across the groups were determined using ANOVA analysis, followed by a Bonferroni post hoc test. The results were expressed as means ± SD of the mean.

## 4. Conclusions

The present study showed that the best strategies for improving the production of *Scutellaria*-specific flavonoids in microshoots of *S. lateriflora* are agitated culture and feeding with the phenylalanine. The same strategy is also favourable for increasing the production of verbascoside. The best condition for total flavonoids and verbascoside production was phenylalanine feeding (1.5 g/L) after 7 days (max. 3765 and 475 mg/100 g DW, respectively). The total flavonoid content was 2.24-fold higher, and the verbascoside content 1.78-fold higher, than the control sample. The results obtained after the combined strategy showed that neither parallel feeding with precursors nor elicitation produced any expected synergistic effect. For the first time, *S. lateriflora* microshoot cultures were maintained in a temporary immersion system (TIS)—Plantform bioreactor, with good biomass increments and metabolite production. Due to the microshoots of *S. lateriflora* being a rich, safe source of antioxidants, further work should focus on the optimization of culture conditions for biomass growth and production of metabolites with high antioxidant activity in TIS bioreactors with higher biotechnological applications, in comparison with agar and agitated cultures (e.g., testing: the chemical composition of culture media, the frequency and time duration of immersion of the biomass with the media during growth cycles, and feeding with precursors and elicitors).

## Figures and Tables

**Figure 1 molecules-27-01140-f001:**
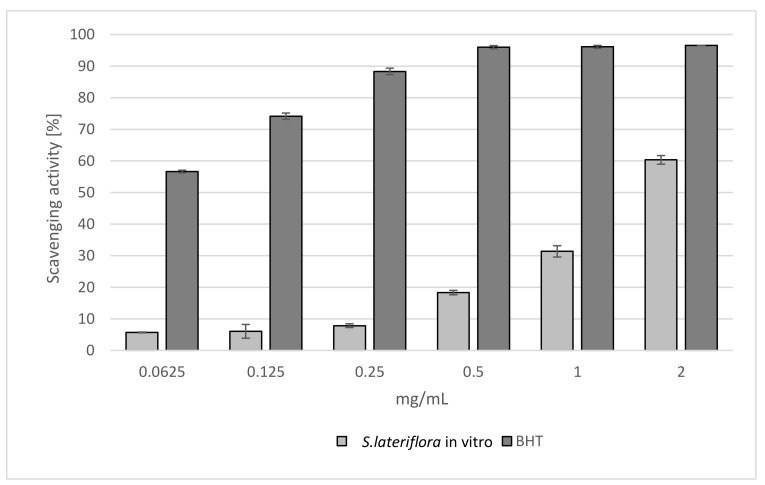
Free radical scavenging activity of the extract obtained from in vitro microshoot cultures of *S. lateriflora* (MS solid medium, supplemented with 1.0 mg/L BA and 0.5 mg/L NAA). Values are expressed as the mean ± SD (*n* = 3).

**Figure 2 molecules-27-01140-f002:**
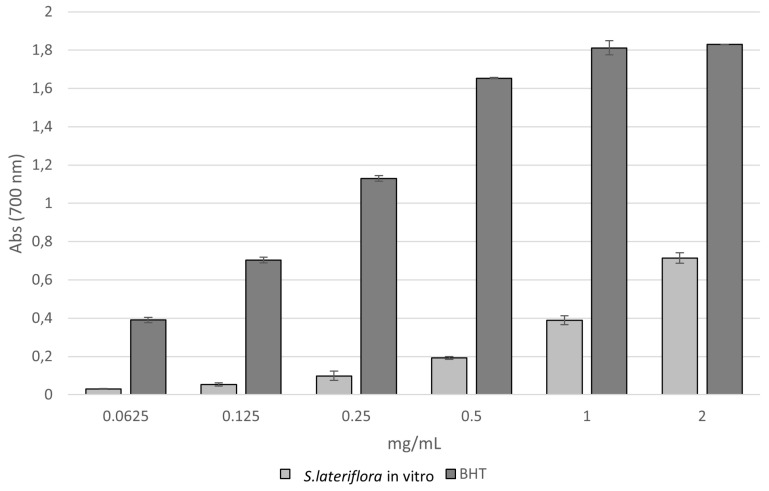
Reducing power of the extract obtained from in vitro microshoot cultures of *S. lateriflora* (MS solid medium, supplemented with 1.0 mg/L BA and 0.5 mg/L NAA) evaluated by spectrophotometric detection of Fe^3+^−Fe^2+^ transformation method. Values are expressed as the mean ± SD (*n* = 3).

**Figure 3 molecules-27-01140-f003:**
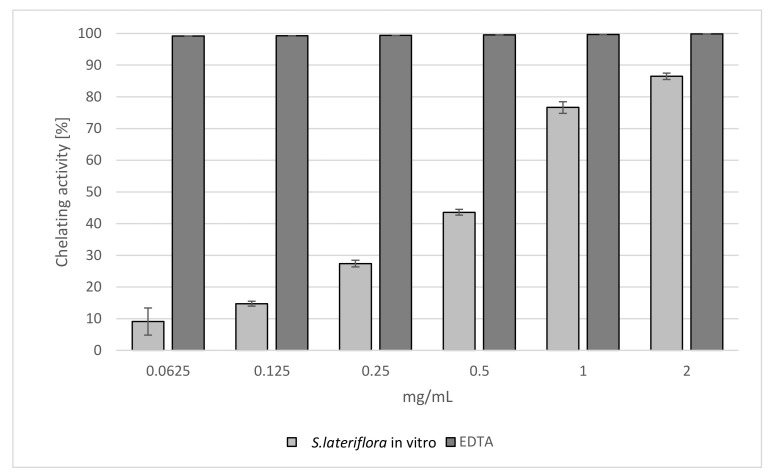
Chelating activity of the extract obtained from in vitro microshoot cultures of *S. lateriflora* (MS solid medium, supplemented with 1.0 mg/L BA and 0.5 mg/L NAA) measured by inhibition of ferrozine-Fe^2+^ complex formation. Values are expressed as the mean ± SD (*n* = 3).

**Figure 4 molecules-27-01140-f004:**
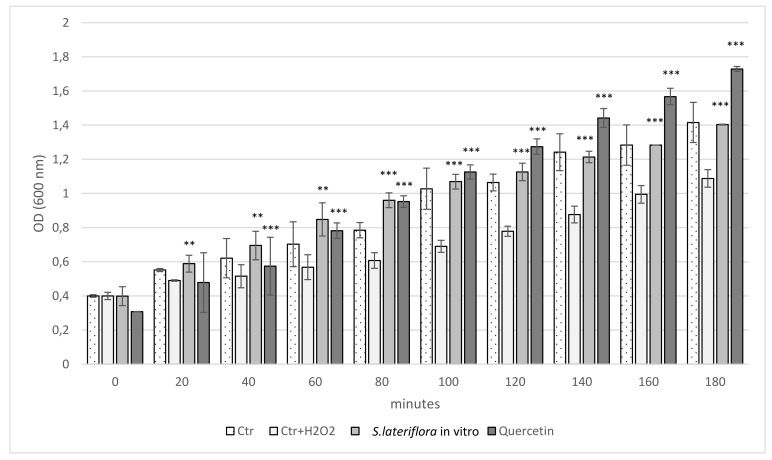
Protective effect of the extract obtained from in vitro microshoot cultures of *S. lateriflora* (MS solid medium, supplemented with 1.0 mg/L BA and 0.5 mg/L NAA) on *E. coli* growth under peroxide stress. Values are expressed as the mean ± SD (*n* = 3). Statistically significant differences compared to control group with H_2_O_2_ treatment (Ctr + H_2_O_2_) are indicated with asterisks (** *p* < 0.01, *** *p* < 0.001).

**Figure 5 molecules-27-01140-f005:**
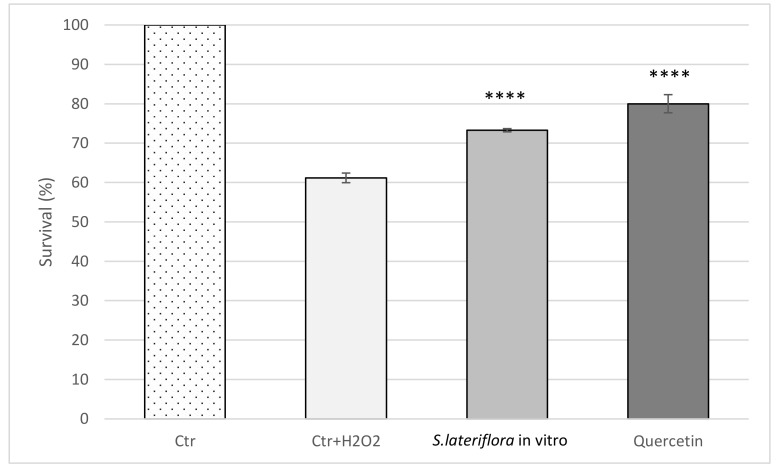
Protective effect of the extract obtained from in vitro microshoot cultures of *S. lateriflora* (MS solid medium, supplemented with 1.0 mg/L BA and 0.5 mg/L NAA) on *E. coli* survival under peroxide stress. Values are expressed as the mean ± SD (*n* = 3). Statistically significant differences compared to control group with H_2_O_2_ treatment (Ctr + H_2_O_2_) are indicated with asterisks (**** *p* < 0.0001).

**Figure 6 molecules-27-01140-f006:**
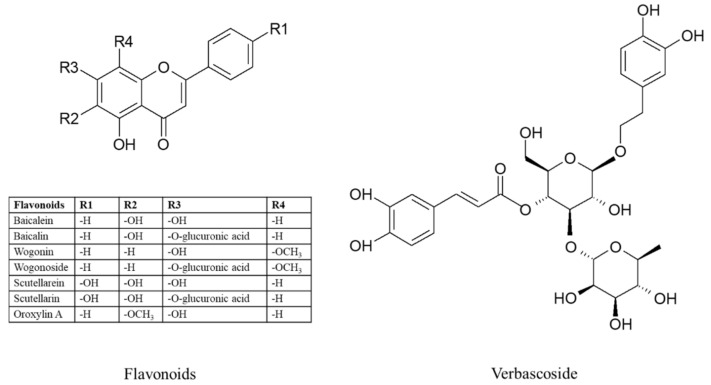
Chemical structure of the compounds detected in the extracts from *S. lateriflora* in vitro cultures.

**Figure 7 molecules-27-01140-f007:**
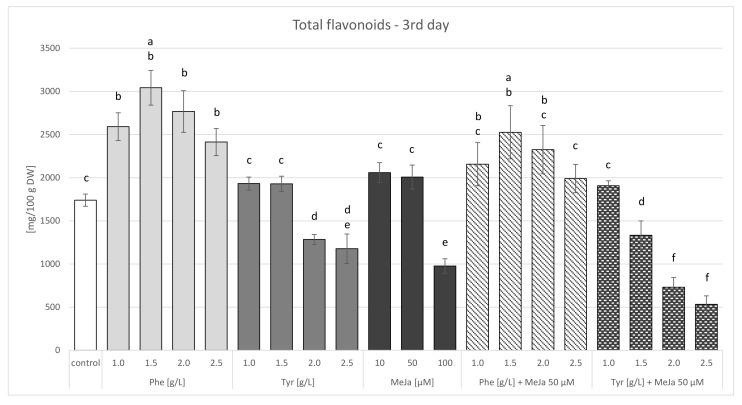
Total flavonoid contents in agitated microshoot in vitro cultures of *S. lateriflora* grown on LS medium supplemented with 1.0 mg/L BA and 1.0 NAA mg/L after administering different concentrations of the biosynthetic precursors—phenylalanine (Phe) and tyrosine (Tyr), and the elicitor—methyl jasmonate (MeJa), collected 3 days after supplementation. ^a–f^ Different letters indicate significant differences (*p* < 0.05).

**Figure 8 molecules-27-01140-f008:**
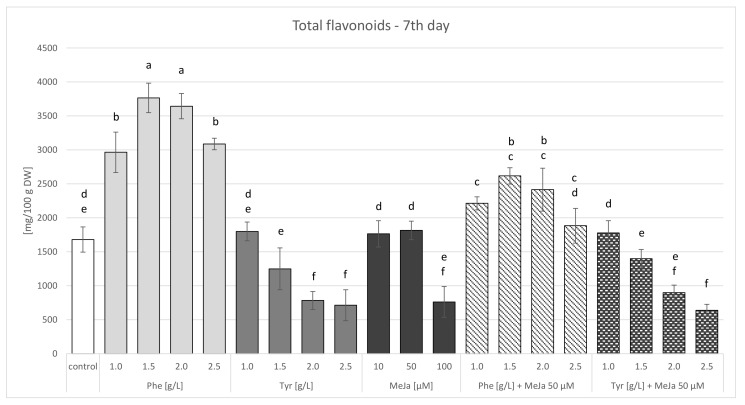
Total flavonoid contents in agitated microshoot in vitro cultures of *S. lateriflora* grown on LS medium supplemented with 1.0 mg/L BA and 1.0 NAA mg/L after administering different concentrations of the biosynthetic precursors—phenylalanine (Phe) and tyrosine (Tyr), and the elicitor—methyl jasmonate (MeJa), collected 7 days after supplementation. ^a–f^ Different letters indicate significant differences (*p* < 0.05).

**Table 1 molecules-27-01140-t001:** Total phenolic content (TPC), total flavonoid content (TFC), free radical scavenging activity (DPPH test), reducing power, and ferrous ion chelating activity of *Scutellaria lateriflora* microshoot extract.

	TPC mg GAE/g Extract	TFC mg QE/g Extract	DPPH IC50 [mg/mL]	Reducing Power ASE/mL	Fe^2+^ Chelating Activity IC50 [mg/mL]
*S. lateriflora* extract	37.270 ± 0.650	8.820 ± 0.900	1.639 ± 0.008 ^a^	43.481 ± 0.237 ^a^	0.609 ± 0.018 ^a^
Reference standard	-	-	BHT 0.066 ± 0.008 ^b^	BHT 1.131 ± 0.008 ^b^	BHT 0.007 ± 0.001 ^b^

Values are expressed as the mean ± SD (*n* = 3). ^a,b^ Different letters within the same column indicate significant differences between mean values (*p* < 0.05).

**Table 2 molecules-27-01140-t002:** Content of phenolic compounds in microshoot in vitro cultures of *S. lateriflora* grown on LS agar medium supplemented with 1.0 mg/L BA and 0.5 mg/L NAA—phenylalanine (Phe) and tyrosine (Tyr) administering [1 g/L].

Metabolite [mg/100 g DW]	Control	Phe	Tyr
Baicalein	239.05 ± 58.668	222.48 ± 49.106	124.56 ± 77.470
Baicalin	1692.15 ± 206.525 ^a^	1658.94 ± 200.664 ^a^	996.90 ± 228.960 ^b^
Wogonin	74.82 ± 43.251 ^a^	95.77 ± 29.820 ^a,b^	35.38 ± 16.405 ^b^
Wogonoside	711.97 ± 4.835 ^a^	491.16 ± 67.224 ^b^	128.29 ± 35.132 ^c^
Scutellarin	4.41 ± 0.270 ^a^	46.79 ± 7.684 ^b^	72.84 ± 21.906 ^b^
Oroxylin A	45.75 ± 19.362 ^a^	14.98 ± 3.887 ^b^	4.41 ± 2.162 ^c^
Total flavonoids	2768.14 ± 332.911 ^a^	2530.13 ± 358.385 ^a^	1262.73 ± 382.035 ^b^
Verbascoside	292.00 ± 65.789 ^a^	457.82 ± 57.807 ^b^	553.72 ± 135.302 ^b^

^a–c^ Different letters indicate significant differences (*p* < 0.05).

**Table 3 molecules-27-01140-t003:** The maximal amounts of estimated metabolites in agitated microshoot in vitro cultures of *S. lateriflora* grown on LS medium supplemented with 1.0 mg/L BA and 1.0 NAA mg/L after administering different concentrations of the biosynthetic precursors phenylalanine (Phe) and tyrosine (Tyr), and the elicitor methyl jasmonate (MeJa), collected 3 days after supplementation.

Metabolite [mg/100 g DW]	3rd Day					
	Control	Phe	Tyr	MeJa	Phe + MeJa	Tyr + MeJa
Baicalein	171.557 ± 12.516	533.084 ± 18.753 ^b^	315.664 ± 28.169 ^b^	351.450 ± 41.801 ^c^	527.640 ± 57.302 ^b^	257.450 ± 8.652 ^a^
Baicalin	445.713 ± 20.473	698.826 ± 35.761 ^b^	442.068 ± 20.460 ^b^	552.318 ± 23.506 ^c^	610.602 ± 78.309 ^b^	507.194 ± 11.708 ^a^
Wogonin	689.250 ± 24.371	808.929 ± 64.103 ^b^	749.526 ± 15.489 ^b^	676.638 ± 25.215 ^d^	745.436 ± 77.691 ^b^	659.016 ± 10.974 ^a^
Wogonoside	424.741 ± 11.473	935.389 ± 67.641 ^b^	407.018 ± 21.356 ^a^	469.450 ± 23.055 ^c^	602.067 ± 85.539 ^b^	442.457 ± 18.941 ^a^
Scutellarin	4.065 ± 0.927	72.340 ± 13.252 ^a^	14.205 ± 1.264 ^b^	25.840 ± 6.515 ^d^	17.999 ± 5.157 ^b^	17.452 ± 1.758 ^a^
Oroxylin A	4.620 ± 0.788	34.814 ± 2.934 ^b^	29.693 ± 4.480 ^a^	37.418 ± 2.325 ^d^	33.493 ± 2.452 ^a^	27.505 ± 2.540 ^b^
Total flavonoids	1739.947 ± 70.548	3041.690 ± 201.914 ^b^	1931.784 ± 75.987 ^a^	2059.550 ± 114.897 ^c^	2525.897 ± 307.879 ^b^	1905.997 ± 55.977 ^a^
Verbascoside	311.197 ± 40.548	469.127 ± 91.914 ^b^	299.480 ± 75.987 ^a^	381.523 ± 114.897 ^c^	160.597 ± 37.879 ^b^	25.460 ± 7.405 ^b^

^a^ 1.0 g/L; ^b^ 1.5 g/L; ^c^ 10 µM; ^d^ 50 µM.

**Table 4 molecules-27-01140-t004:** The maximal amounts of estimated metabolites in agitated microshoot in vitro cultures of *S. lateriflora* grown on LS medium supplemented with 1.0 mg/L BA and 1.0 NAA mg/L after administering different concentrations of the biosynthetic precursors phenylalanine (Phe) and tyrosine (Tyr), and the elicitor methyl jasmonate (MeJa), collected 7 days after supplementation.

Metabolite [mg/100 g DW]	7th Day					
	Control	Phe	Tyr	MeJa	Phe + MeJa	Tyr + MeJa
Baicalein	162.414 ± 26.104	614.788 ± 38.279 ^b^	162.457 ± 49.878 ^b^	300.486 ± 40.901 ^e^	541.608 ± 36.215 ^b^	264.457 ± 31.590 ^a^
Baicalin	378.595 ± 26.606	988.086 ± 56.083 ^c^	419.209 ± 21.780 ^a^	474.399 ± 49.675 ^d^	630.484 ± 17.532 ^b^	411.619 ± 41.777 ^a^
Wogonin	666.088 ± 71.518	964.501 ± 49.668 ^b^	655.544 ± 30.779 ^a^	676.857 ± 55.040 ^e^	782.986 ± 29.100 ^b^	599.387 ± 51.315 ^a^
Wogonoside	469.987 ± 59.996	1148.771 ± 50.697 ^c^	516.879 ± 56.240 ^a^	365.481 ± 17.298 ^e^	620.747 ± 30.154 ^b^	479.433 ± 51.315 ^a^
Scutellarin	2.539 ± 0.940	68.295 ± 4.980 ^b^	9.321 ± 1.797 ^a^	10.087 ± 2.591 ^e^	18.946 ± 3.460 ^a^	10.042 ± 1.025 ^b^
Oroxylin A	0.214 ± 0.023	53.653 ± 4.450 ^c^	41.758 ± 8.250 ^b^	43.708 ± 3.768 ^d^	25.006 ± 7.145 ^c^	14.562 ± 2.246 ^a^
Total flavonoids	1679.837 ± 185.187	3764.881 ± 217.489 ^b^	1799.470 ± 137.499 ^a^	1814.878 ± 137.456 ^e^	2616.874 ± 120.568 ^b^	1778.499 ±179.168 ^a^
Verbascoside	267.327 ± 15.187	474.795 ± 27.489 ^b^	263.961 ± 37.499 ^a^	121.553 ± 19.489 ^d^	170.694 ± 17.156 ^a^	44.068 ± 4.892 ^b^

^a^ 1.0 g/L; ^b^ 1.5 g/L; ^c^ 2.0 g/L; ^d^ 10 µM; ^e^ 50 µM.

**Table 5 molecules-27-01140-t005:** Content of phenolic compounds in microshoot in vitro cultures of *S. lateriflora* grown in Plantform bioreactors on two types of medium—MS medium supplemented with 1.0 mg/L BA and 0.5 NAA mg/L, and LS medium supplemented with 1.0 mg/L BA and 0.5 NAA mg/L.

Metabolite * [mg/100 g DW]	BA/NAA [mg/L]	
	MS 1.0/0.5	LS 1.0/0.5
Baicalin	1388.74 ± 140.169 ^a^	2191.23 ±399.847 ^b^
Wogonin	169.22 ± 48.145 ^a^	68.56 ± 13.553 ^b^
Wogonoside	174.99 ± 61.832 ^a^	53.67 ± 21.344 ^b^
Scutellarin	10.55 ± 3669	16.91 ± 5358
Oroxylin A	39.87 ± 3835 ^a^	17.49 ± 8345 ^b^
Total flavonoids	1783.38 ± 257.651	2347.87 ± 448.447
Verbascoside	485.45 ± 47.439 ^a^	310.37 ± 37.778 ^b^

* The presence of baicalein was not confirmed; ^a,b^ different letters indicate significant differences (*p* < 0.05).

## Data Availability

Not applicable.

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
