# Peer review of "Antioxidant Potential and Enhancement of Bioactive Metabolite Production in In Vitro Cultures of *Scutellaria lateriflora* L. by Biotechnological Methods"

_molecules, 2022, doi:10.3390/molecules27031140_

Round 1
Reviewer 1 Report
The present study investigate the antioxidant potential and enhancement of bioactive metabolites production in in vitro cultures of Scutellaria lateriflora L. by biotechnological methods. The modification suggestions are shown below:
- Figure 1,2,3,6,7 and table 1,4 no significance analysis.
- The discussion should be discussed in depth in combination with existing literature reports.
- According to the sentence “In vitro and in vivo studies (4 different methods), demonstrated antioxidant activity of 12 Scutellaria lateriflora microshoots extract.” Which part of the experiment reflects the in vivo experiment?
- What is the relationship between improving the production of Scu-704 tellaria-specific flavonoids in microshoots of S. lateriflora and the antioxidant activity? How about the bioactivities of metabolites?
- We recommend providing a diagram of the experimental process.
- This paper lacks a discussion part, only the exposition of the experimental process and results, without a deep discussion.
Reviewer 2 Report
The manuscript suggest adequately the species and proccesses to expand antioxidants from Scutellaria lateriflora. Some points should be edited for improving presentation.
a) Please check results for avoiding repetitive presentation of results.
b) The chemical structure of some mentioned molecules is desirable.
c) The discussion is poor regarding the putative differences in relationship to structural features. Also, discussion of the differences when Phe or Tyr were used as precursors should be extended.
d) Limitations should be sentenced, among these, the specific identification of some flavonoids and phenols by using spectrometry assays should be mentioned.
e) Conclusions must be clear and centered in those findings on the present manuscript. But some comparisson of similar assays with other processes and species could be added. Also, a prospective sentence at the end could improve the expanding condition in the field.
f) Please check the adequate use of italics letters.
Reviewer 3 Report
This review describes the different biotechnological approaches for improving bioproduction of potentially bioactive secondary metabolites in American skullcap (Scutellaria lateriflora L.). The topic of this work is interesting and relevant. It may be speculated that it will contribute to the development of current knowledge in this field. In general, the introduction is quite good described (some details are described in comments). Modes of results presentation are clear. Discussion and conclusions are properly described and supported by the results. However, some clarifications in methodology are needed. It is a lack of statistical analysis for some results. In addition, experiment design could be better planned. It is very difficult to follow the trajectory of the chosen analysis, treatments, and consequently obtained results. Experiment design and results presentation it is very inconsistent and incompatible. It has to be ordered. It is a lack of presentation of some described results (in the form of figures or tables).
In the case of in vitro micro shoot cultures theoretically, the total phenolics and flavonoids contents were determined (lack of results – Fig. or Tab.). Additionally, antioxidant activities: DPPH radical scavenging activity, reducing power, chelating ability, protective effect on E. coli growth under peroxide stress, and Artemia salina Lethality Bioassay( (lack of results – Fig. or Tab.), were determined. However, the effect of precursor feeding, elicitation and combined strategy was not analyzed, as in the case of agitated cultures. Why were not performed HPLC analysis as in the case of agar, agitated and bioreactor cultures?
For agar cultures, only the phenolic profile (HPLC) was determined. Phenolic profile (HPLC) was analyzed as affected by precursor feeding (only). Why was analyzed only precursor feeding analyzed, but not also elicitation and combined strategy, as for agitated cultures? Why were not performed TPC, TFC, and antioxidant assays, as for in vitro micro shoot cultures?
For agitated cultures, total flavonoid content or selected flavonoids contents (HPLC) after 3 or 7 days of supplementation were determined, as affected by precursor feeding, elicitation and combined strategy. Why were not studied antioxidant activities? Why were not performed TPC, TFC, and antioxidant assays, as for in vitro micro shoot cultures?
For bioreactor cultures, only the effect of different medium on the HPLC profile of compounds was analyzed. However, antioxidant activities, TFC, TPC, were not studied. Furthermore, the effect of precursor feeding, elicitation and combined strategy was not analyzed.
According to the reviewer, knowledge Molecules has no restrictions on the length of manuscripts and in the number of tables and figures. So, additional results can be added to obtain consequence and consistency through the manuscript. Conversely, if it is not possible, this work should be divided into more than one. The results can be divided and presented separately e.g. in three different 3 manuscripts, where authors should focus only on the one cultivation method. Presented hotchpotch significantly decrease the scientific value of this manuscript and in its current form can’t be accepted. It needs substantial revision.
Detailed revisions:
Abstract
Line 23 – „antioxidant production” rather antioxidants production
Introduction
Line 60 – 77 – The previous authors' achievements (of course valuable) seem not to be very important as the background of considerations of current work, but they can be implemented in the discussion section (e.g. results comparison with previous studies). I recommended removing it from the introduction.
Line 87-89 – „Aromatic amino acids, namely phenylalanine and tyrosine, were used as precursors of phenolic compounds. Methyl jasmonate, a biotic elicitor, was used in the elicitation experiment”
It will be valuable to develop this issue by the justification of these compounds selection, (e.g. justify the application of phenylalanine and tyrosine as precursors, regarding polyphenols biosynthesis pathways)
Results and discussion
Line 101-106 – add results, why they are not presented?
Line 123 – „As indicated by IC50 values the activity of the extract was found lower than that of butylated hydroxytoluene (BHT) used as a standard 124 drug (IC50= 1.639 ± 0.008 mg/mL and 0.066 ± 0.008 mg/mL, respectively) (Figure 1).” Figure 1 presents only % scavenging activity. Add IC50. Fig. 1 - Why is 0.5 mg/ml BHT isn’t treated as 100% activity? It is clearly shown that higher conc. of BHT didn’t decolourize DPPH – it means 100% activity was achieved.
Line 134 – where are (ASE)/mL values depicted in Fig.?
Line 148 – Fig. 3 is cut. IC 50 – see above comments.
Table 1, 4, supplementary – add statistical analysis (eg. SD and statistical differences indicated using letters). Add SD to other tables.
Table 2 and 4 - why the statistical analysis was not performed for control?
Materials and methods
Line 563 – „After cultivation, the biomass was harvested and dried at room temperature.” Why this method of drying was applied? Please, justify. In my opinion, it would be better to apply drying under controlled conditions. eg. convection drying in mild temperature e.g. 30C – 40C. Room temperature drying in a “natural way” is in general long-lasting. During long-lasting drying, polyphenol oxidases, especially at the initial step of drying can lead to the degradation of polyphenols. In many cases drying at a slightly higher temperature (but also low) can speed up the process and reduce the mentioned effect. Lyophilization could be also considered.
Line 584 – add information about downstream processing i.e. samples preparation for extraction (e.g. drying method). Authors should consider the bioactive compounds excretion to the medium during bioreactor cultivation (in agitated and agar cultures, as well).
Line 556 – “We chose this medium as our previous experiments documented that it was the best “productive” medium for this type of in vitro culture [20]. Add at the end of this phrase „ among tested.”
Line 600-602- „(27 flavonoids, 19 phenolic acids, benzoic and cinnamic acids, phenylethanoid glycosides - verbascoside and isoverbascoside) (Chro-601 maDex, Irvine, USA; Sigma–Aldrich Co., Saint Louis, USA; ChemFaces Biochemical Co., 602 Wuhan, PRC).” Please remove this unnecessary information, because not all of these compounds were not analyzed in this study. The next phrase gives specific information for this study.
Line 607 – remove part of a paragraph concerning total phenolic content ( results description in text) if results are not shown in the manuscript (Tab. or Fig.) or add results
Line 644 – It will be better to prepare a standard curve for BHT and express results as equivalents of BHT (or Trolox) per DW of plant material. „The results averaged from three independent experiments were expressed as mean radical scavenging activity (%) ± SD and mean 50% inhibitory concentration (IC50) ± SD” IC 50 values should be presented at Fig. instead % of activity. Correct it.
Line 661 – where are the results for ascorbic acid (Fig. 2)? Why results at Fig. 2. are expressed as absorbance instead of as ascorbic acid equivalents (ASE/mL) (line 663)
Line 672-674 – Where are the results for EDTA and values of IC 50. In Fig. 3 results are expressed as % chelating activity. Correct it.
Statistical analysis is not described in this section.
Round 2
Reviewer 1 Report
The revised manuscript can be accpected.
Reviewer 3 Report
Authors should pay more attention to research design in such type of work. The manuscript presents a bit of concept confusion. However, some provided clarifications and partly improvement of the manuscript (compared to the previous version) allow meeting the basic requirements to be published.